# Targeted Hybridization Capture of SARS-CoV-2 and Metagenomics Enables Genetic Variant Discovery and Nasal Microbiome Insights

Dorottya Nagy-Szakal,[a,b] Mara Couto-Rodriguez,[a] Heather L. Wells,[a] Joseph E. Barrows,[a] Marilyne Debieu,[a] Kristin Butcher,[c] Siyuan Chen,[c] Agnes Berki,[d] Courteny Hager,[a] Robert J. Boorstein,[e] Mariah K. Taylor,[f] ⓘ Colleen B. Jonsson,[f] Christopher E. Mason,[a,g,h,i,j] ⓘ Niamh B. O'Hara[a,b,k]

[a]Biotia, Inc., New York, New York, USA
[b]Department of Cell Biology, College of Medicine, SUNY Downstate Health Sciences University, New York, New York, USA
[c]Twist Bioscience, South San Francisco, California, USA
[d]School of Natural Sciences, College of Natural, Behavioral and Health Sciences, Caldwell University, Caldwell, New Jersey, USA
[e]Lenco Diagnostic Laboratories, Inc., New York, New York, USA
[f]The University of Tennessee Health Science Center, Memphis, Tennessee, USA
[g]Tri-Institutional Computational Biology & Medicine Program, Weill Cornell Medicine, Cornell University, New York, New York, USA
[h]HRH Prince Alwaleed Bin Talal Bin Abdulaziz Alsaud Institute for Computational Biomedicine, Weill Cornell Medicine, New York, New York, USA
[i]WorldQuant Initiative for Quantitative Prediction, Weill Cornell Medicine, New York, New York, USA
[j]Feil Family Brain and Mind Research Institute, Weill Cornell Medicine, New York, New York, USA
[k]Jacobs Technion-Cornell Institute, Cornell Tech, New York, New York, USA

Dorottya Nagy-Szakal and Mara Couto-Rodriguez contributed equally to this article. Author order was determined on the basis of seniority.

**ABSTRACT** The emergence of novel severe acute respiratory syndrome coronavirus 2 (SARS-CoV-2) genetic variants that may alter viral fitness highlights the urgency of widespread next-generation sequencing (NGS) surveillance. To profile genetic variants of the entire SARS-CoV-2 genome, we developed and clinically validated a hybridization capture SARS-CoV-2 NGS assay, integrating novel methods for panel design using double-stranded DNA (dsDNA) biotin-labeled probes, and built accompanying software. This test is the first hybrid capture-based NGS assay given Food and Drug Administration (FDA) emergency use authorization for detection of the SARS-CoV-2 virus. The positive and negative percent agreement (PPA and NPA, respectively) were defined in comparison to the results for an orthogonal real-time reverse transcription polymerase chain reaction (RT-PCR) assay (PPA and NPA, 96.7 and 100%, respectively). The limit of detection was established to be 800 copies/ml with an average fold enrichment of 46,791. Furthermore, utilizing the research-use-only analysis to profile the variants, we identified 55 novel mutations, including 11 in the functionally important spike protein. Finally, we profiled the full nasopharyngeal microbiome using metagenomics and found overrepresentation of 7 taxa and evidence of macrolide resistance in SARS-CoV-2-positive patients. This hybrid capture NGS assay, coupled with optimized software, is a powerful approach to detect and comprehensively map SARS-CoV-2 genetic variants for tracking viral evolution and guiding vaccine updates.

**IMPORTANCE** This is the first FDA emergency-use-authorized hybridization capture-based next-generation sequencing (NGS) assay to detect the SARS-CoV-2 genome. Viral metagenomics and the novel hybrid capture NGS-based assay, along with its research-use-only analysis, can provide important genetic insights into SARS-CoV-2 and other emerging pathogens and improve surveillance and early detection, potentially preventing or mitigating new outbreaks. Better understanding of the continuously evolving SARS-CoV-2 viral genome and the impact of genetic variants may provide individual risk stratification,

Address correspondence to Dorottya Nagy-Szakal, dorottya.nagy-szakal@downstate.edu, or Niamh B. O'Hara, niamh.ohara@cornell.edu.

precision therapeutic options, improved molecular diagnostics, and population-based thera-peutic solutions.

**KEYWORDS** COVID-19, SARS-CoV-2, infectious disease, microbiome, next-generation sequencing, viral genomics

The coronavirus disease 2019 (COVID-19) pandemic has resulted in an unprecedented disruption of life across the globe and an unparalleled death toll (1, 2). To guide pub-lic health decisions and control the pandemic, a number of tests for the detection of severe acute respiratory syndrome coronavirus 2 (SARS-CoV-2) have been developed and implemented globally (3). Despite reverse transcription polymerase chain reaction (RT-PCR) being rapid and inexpensive, it does not characterize the virus and is prone to failure when viruses acquire mutations (4). As functional genetic variants emerge, there is a vital need for sequencing-based approaches that tolerate mutations and characterize the viral genome (5). Direct metagenomic sequencing does not require prior knowledge of the genomic sequence but has limited sensitivity due to the small viral genome and low abundance of the virus compared to the high background from the human host and other microbes. Without deep sequencing, which is costly, shotgun sequencing approaches may miss genetic variants due to low or variegated coverage. A targeted approach, which enriches for specific sequences out of a mixed genomic sample, over-comes these difficulties, improving the sensitivity and specificity and achieving higher coverage across the genome (6). The widely used amplicon-based approach (e.g., ARTIC [7]) uses many pairs of overlapping primers to amplify viral genome prior to sequencing. However, it has limited tolerance of mismatches between the targeted sequences and the primers that are used, resulting in an increased risk of amplicon failure as the virus continues to evolve. The hybrid capture-based method relies on hybridization between the target genome and relatively long probes (up to 120 bp) that are designed to be comple-mentary to the target genome. The method can tolerate large target sequence differences of $\sim$10% or more from the probe sequences (8, 9). The higher tolerance for mismatches makes the successful enrichment of highly divergent SARS-CoV-2 sequences possible (10).

Public databases, such as GISAID and SPHERES (SARS-CoV-2 Sequencing for Public Health Emergency Response, Epidemiology and Surveillance), have been collecting sequence data to provide global genetic surveillance (11, 12). However, using mostly amplicon-based next-generation sequencing (NGS) data may have resulted in missed variants due to primer failures or insufficient sequencing depth. There is an emerging need for the implementation of genomic tools to generate robust and accurate sequence data to assess circulating variants, and the United States is lagging behind many other countries while also carrying more cases (2).

As we expand sequencing efforts, it is also of interest to ascertain if infection by SARS-CoV-2 may be associated with an altered respiratory tract microbiome and influence dis-ease outcome or if patients with specific microbial profiles may be more susceptible to SARS-CoV-2 infection (13). Overrepresentation of not only respiratory pathogens but also opportunistic respiratory microbes (such as *Prevotella* spp.) could affect the inflammatory response and enhance the disease severity (14). Understanding the microbiome of the re-spiratory tract, potential pathogens, virulence factors (VFs), and antimicrobial resistance (AMR) in SARS-CoV-2-positive patients may elucidate the pathogenicity of COVID-19 in relation to the microbiome and other respiratory pathogens. Characterization of the role and rise of AMR is additionally underscored given the dramatic increase in the use of anti-biotics to treat pneumonia during the pandemic.

Here, we developed and clinically validated the SARS-CoV-2 NGS assay and COVID-DX software, the first hybridization capture-based SARS-CoV-2 NGS test given emergency use authorization by the FDA with a research-use-only option for highly sensitive characterization of genetic variants (Fig. 1). Additionally, we profiled the nasopharyngeal (NP) microbiome to allow greater understanding of coinfections and drug resistance that contribute to the mor-bidity and mortality caused by SARS-CoV-2.

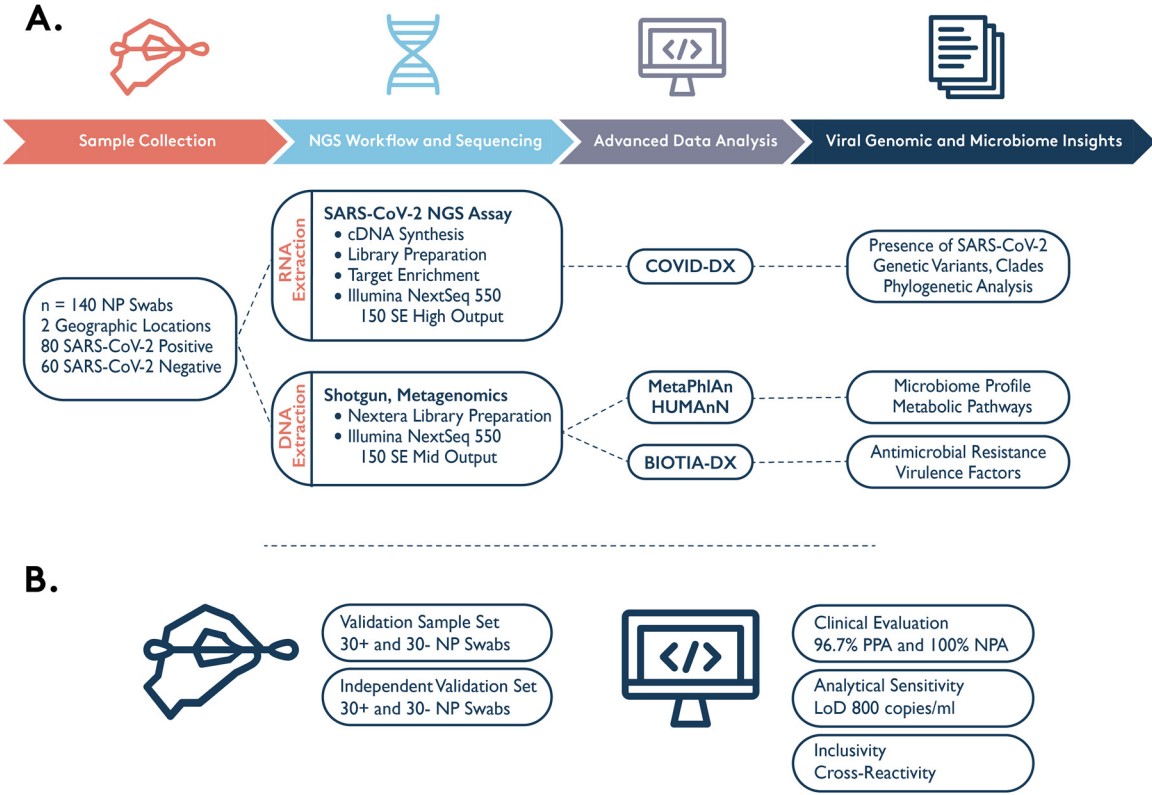

**FIG 1** Schematic description of the study workflow (A) and validation (B).

## RESULTS

**Clinical performance and analytical sensitivity.** The clinical performance of the SARS-CoV-2 NGS assay was evaluated by comparing results to those of real-time RT-PCR assay approved by the U.S. Food and Drug Administration (FDA) through an emergency use authorization (EUA) authority. A total of 60 clinical NP specimens were tested, including 30 SARS-CoV-2-positive (CVP) and 30 SARS-CoV-2-negative (CVN) specimens. The positive and negative percent agreement (PPA and NPA, respectively) were calculated in relation to the real-time RT-PCR comparator method and resulted in 96.7% PPA and 100% NPA. Additional independent clinical validation (an additional 30 CVP and 30 CVN) resulted in 93.3% PPA and 100% NPA (Table 1 and Table S2 in the supplemental material). Subsampling to 500,000 reads per sample did not significantly change the fold enrichment, on-target reads, or presence calling of SARS-CoV-2 viral genome (Table S2).

The analytical sensitivity established the lowest concentration of SARS-CoV-2 detected by our assay at least 95% of the time to be a limit of detection (LOD) of 800 copies/ml (0.8 copies/$\mu$l), which is better than the LOD of several dozen other FDA EUA assays (15). Results of the 2-fold and 10-fold LOD dilution of synthetic and inactivated viral controls can be found in Table S3.

Our hybrid capture technology yielded an average of 43% on-target reads (ranging from 0.005% to 99.5%) and an average fold enrichment of 46,791 (ranging from 5.9 to 108,602) based on the alignment statistics.

To better understand NGS assay performance, cycle threshold ($C_T$) values were obtained for 44 CVP specimens by running an additional real-time RT-PCR assay. Figure 2 and Tables S1 and S2 show the fold enrichment, depth of coverage, and percent coverage of the genome at different depths in relation to $C_T$ values.

**Inclusivity and exclusivity studies.** In our inclusivity analysis, we evaluated six synthetic controls and the novel United Kingdom and South African strains and confirmed the assay's ability to capture the different SARS-CoV-2 lineages. We additionally performed

**TABLE 1** Performance of clinical evaluation[a]

| SARS-CoV-2 NGS assay | Result | EUA RT-PCR comparator assay | | % agreement (95% CI) |
| | | Positive | Negative | |
|---|---|---|---|---|
| Validation set | Positive | 29 | 0 | PPA, 96.7 (83.3–99.4) |
| | Negative | 1 | 29 | NPA, 100 (88.3–100) |
| | Invalid | 0 | 1[b] | |
| | | | | |
| Independent validation set | Positive | 28 | 0 | PPA, 93.3 (78.7–98.2) |
| | Negative | 2 | 30 | NPA, 100 (88.6–99.1) |
| | Invalid | 0 | 0 | |

[a]Clinical performance of the SARS-CoV-2 NGS assay was evaluated by comparing results to real-time RT-PCR assay authorized by the FDA for use under emergency use authorization (EUA RT-PCR). Sixty clinical nasopharyngeal swab specimens were tested, including 30 SARS-CoV-2-positive and 30 SARS-CoV-2-negative specimens. The positive percent agreement (PPA) and negative percent agreement (NPA) were calculated in relation to the EUA RT-PCR comparator method. The performance characteristics of an additional independent validation set are also shown. 95% CI, 95% confidence interval.
[b]One PCR-negative sample did not yield sufficient reads using the SARS-CoV-2 NGS assay to be called negative and was labeled invalid.

an extensive *in silico* inclusivity analysis using the GISAID database, identifying 151,323 unique sequences with high identity matches (Table S4). Given the high identity for the majority of probes in the majority of sequences, we expect to have a high probability of identifying the presence or absence of most known strains of SARS-CoV-2.

To account for potential cross-reactivity of the SARS-CoV-2 NGS hybridization probes, we aligned them with 30 microbial genomes, the human genome, and the SARS-CoV-2 genome (Table S4). All pathogens were determined to have no cross-reactivity (no more than 80% homology) with the probes used, except for human coronavirus HKU1 and SARS-CoV-1. These cross-reactive probes are of little clinical concern due to low probe coverage and/or low infection rates by these organisms currently. The *in silico* exclusivity analysis containing approximately 3.6 million viral sequences from NCBI Virus showed no cross-reactivity (Table S4), with the exception of high homology regions of some closely conserved viruses, such as bat coronavirus, infectious bronchitis, and transmissible gastroenteritis viral strains.

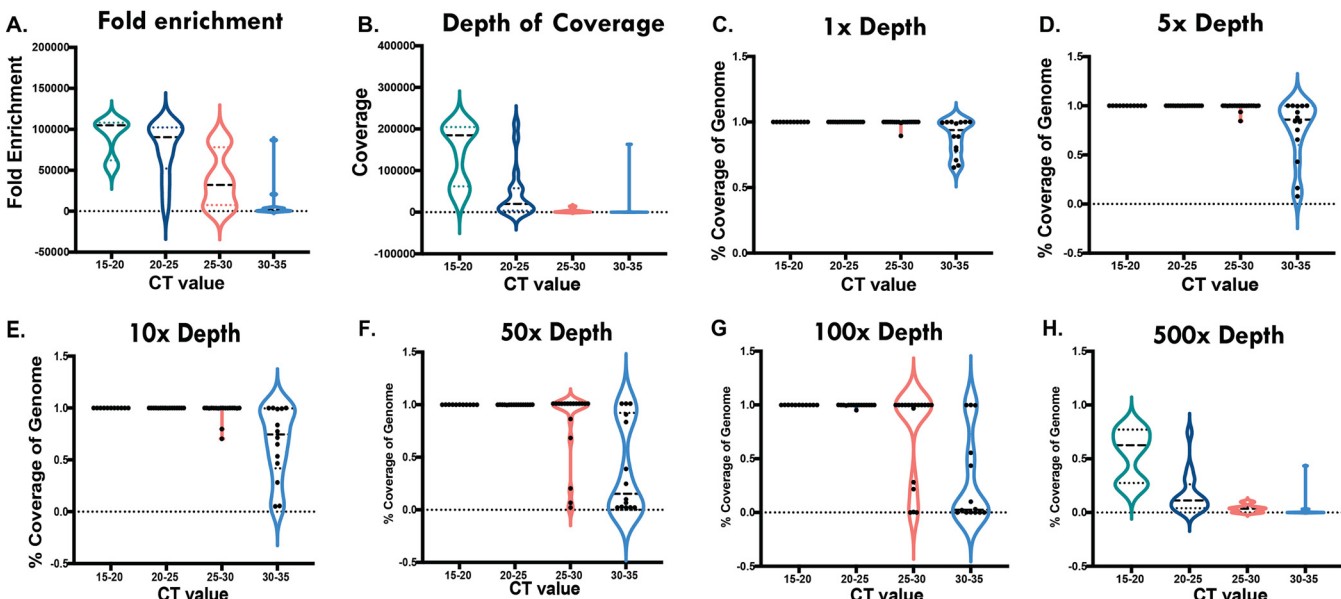

**FIG 2** SARS-CoV-2 hybrid capture consistently provides ample fold enrichment, depth of coverage and percent coverage of genome at various depths in samples with $C_T$ values between 15 and 30. Violin plots depict the sample distributions of fold enrichment (A), depth of coverage (B), and percent coverage of the genome at depths of 1× (C), 5× (D), 10× (E), 50× (F), 100× (G), and 500× (H) in different $C_T$ value groups. Quartiles are depicted by dotted lines, and median values are shown by dashed lines. Results confirm that our samples captured a wide range of viral loads, including $C_T$ values of 15 to 20 (18.6%, $n = 11/59$), 20.5 to 25 (27.1%, $n = 16/59$), 25.5 to 30 (30.5%, $n = 18/59$), and 30.5 to 35 (23.7%, $n = 14/59$).

**TABLE 2** List of the most frequent mutations in all samples and those unique for each geographic location[a]

| Nucleotide mutation | Amino acid mutation | Geo | Samples with mutation | GISAID events | Gene | Protein | Synonymity | Amino acid alteration property |
|---|---|---|---|---|---|---|---|---|
| | | | No. of: | | | | | |
| Most frequent[b] | | | | | | | | |
| C241T | Extragenic | All | 65 | 7 | 5′ UTR | NA | NA | NA |
| A23403G | D614G | All | 65 | 6 | S gene | Spike | Non-S | Radical |
| C14408T | P4715L | All | 62 | 8 | ORF1ab | NSP12/RNApol | Non-S | Radical |
| C3037T | F924F | All | 59 | 10 | ORF1ab | NSP3 | S | NA |
| G25563T | Q57H | All | 59 | 9 | ORF3a | ORF3a | Non-S | Conservative |
| C1059T | T265I | All | 56 | 13 | ORF1ab | NSP2 | Non-S | Conservative |
| G29540A | Extragenic | All | 16 | 2 | N/ORF10 linking region | NA | NA | NA |
| C11916T | S3884L | All | 15 | 4 | ORF1ab | NSP7/replicase | Non-S | Radical |
| C18998T | A6245V | All | 15 | 1 | ORF1ab | NSP14/NSP11 | Non-S | Conservative |
| | | | | | | | | |
| Unique[c] | | | | | | | | |
| G29540A | Extragenic | NYC | 16 | 2 | N/ORF10 linking region | NA | NA | NA |
| C11916T | S3884L | NYC | 15 | 4 | ORF1ab | NSP7/replicase | Non-S | Radical |
| C18998T | A6245V | NYC | 15 | 1 | ORF1ab | NSP14/NSP11 | Non-S | Conservative |
| C4113T | A1283V | NYC | 4 | 6 | ORF1ab | NSP3 | Non-S | Conservative |
| C27964T | S24L | TN | 7 | 7 | ORF8 | ORF8 | Non-S | Radical |
| C3411T | A1049V | TN | 7 | 3 | ORF1ab | NSP3 | Non-S | Conservative |
| T6394C | D2043D | TN | 7 | 3 | ORF1ab | NSP3 | S | NA |
| C10319T | L3352F | TN | 7 | 4 | ORF1ab | NSP5 | Non-S | Conservative |
| A4197G | E1311G | TN | 4 | 2 | ORF1ab | NSP3 | Non-S | Radical |
| G8179A | R2638R | TN | 4 | 4 | ORF1ab | NSP3 | S | NA |
| C15924T | Y5220Y | TN | 4 | 1 | ORF1ab | NSP12/RNApol | S | NA |

[a]Geo, geographic location; UTR, untranslated region; NA, not applicable; Non-S, nonsynonymous; S, synonymous; NSP, nonstructural protein; RNApol, RNA polymerase; NYC, New York City; TN, Tennessee.
[b]Most frequent mutations in all samples.
[c]Unique mutations for each geographic location.

**Novel genetic variants were detected using the hybrid capture NGS-based approach.** After validation of our assay, we defined 72 CVP samples that would undergo further genetic variant analysis and compared these against the globally available database (GISAID). We detected 641 mutations at 179 different mutation sites (Table S5), of which the most frequently detected variants were a change of C to T at position 241 (C241T) (extragenic), A23403G (encoding a change of D to G at position 614 [D614G]), C14408T (P4715L), C3037T (F924F), G25563T (Q57H), and C1059T (T265I), detected in more than 55 samples (Table 2 and Fig. 3A and B). The majority of mutations were located in the ORF1ab gene (64%, 115/179) and in the S gene (12%, 21/179). Altogether, we found 118 nonsynonymous mutations, 5 frameshifts after synonymous mutations, and 37 radical mutations. As of 11 November 2020, we had identified 55 new mutations that have not been previously described in GISAID (Fig. 3C), with the majority located in the ORF1ab gene (37/55, 67%) and the S gene (11/55, 20%), including 42 nonsynonymous mutations, 10 radical mutations, and 2 frameshifts after synonymous mutations. In a recent analysis (23 June 2021) of the mutations detected with our assay, we found 38 mutations that still have not been reported in GISAID. Our validation of the variant calling confirmed selected variants that are highlighted in Table S6.

Samples collected from two geographic locations (New York City and Tennessee) shared several similar genetic variants. The unique variants identified in the New York City and Tennessee samples are listed in Table 2.

**Phylogenetic and geographic origin (clade) results.** We next performed a phylogenetic analysis to provide additional information on the origins and diversity of the SARS-CoV-2 viral genomes based on genetic variants detected in the clinical samples. We identified clades 19A, 20A, and 20C as being the most abundant clades ($n$ = 14, 23, and 27, respectively) (Fig. 3D and Table S1), in concordance with observations of New York cohorts (5).

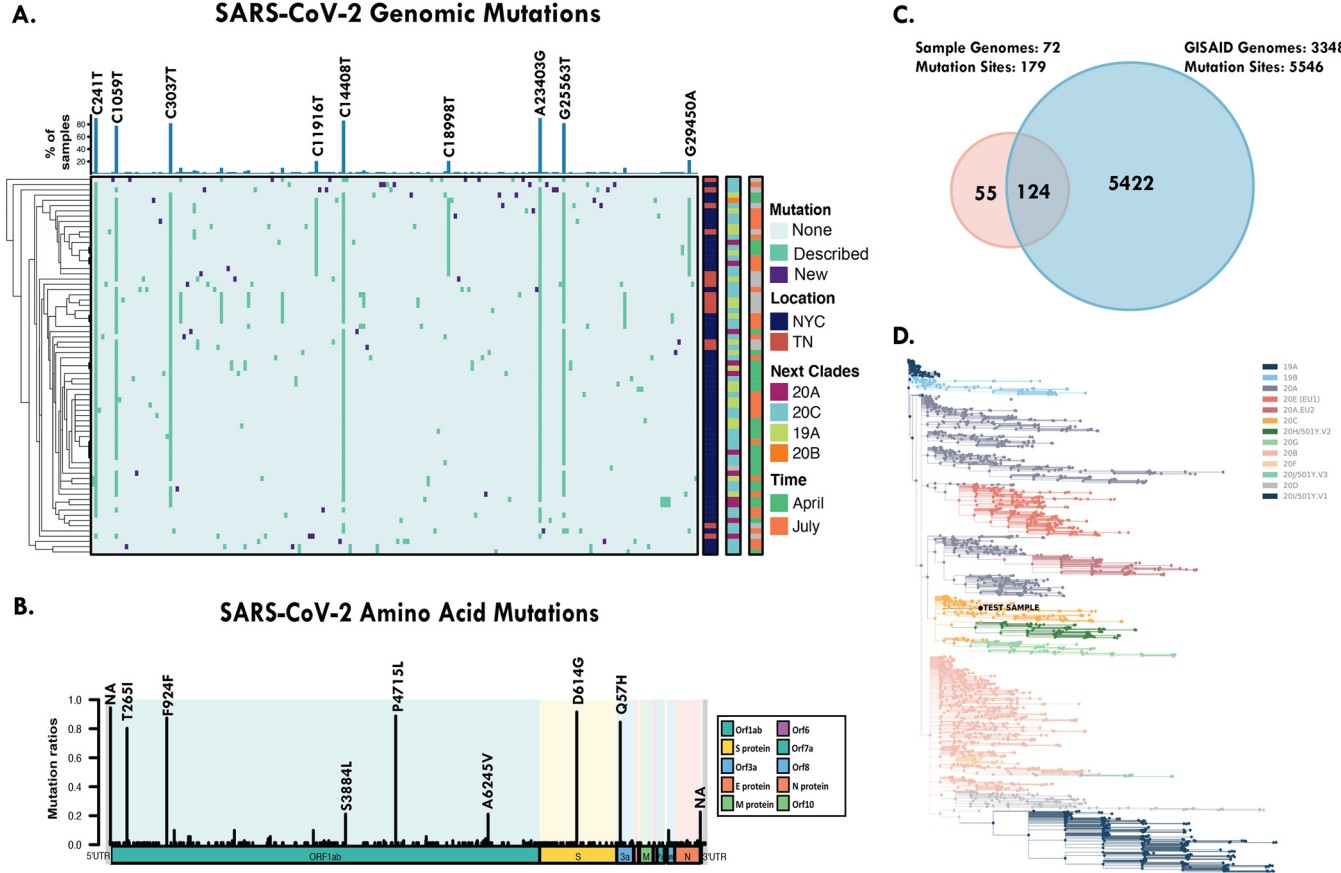

**FIG 3** SARS-CoV-2 hybrid capture panel identified novel genetic variants. (A) Heatmap depiction of the nucleotide mutations detected within our SARS-CoV-2 cohort (n = 72). Top bar plot annotations indicate the percentages of samples containing nucleotide substitutions; those labeled are the most prevalent. Annotations to the right of the heatmap indicate the presence or absence of mutations (new mutations, previously described mutations, and no mutations compared to the SARS-CoV-2 Wuhan strain), geographic location (New York City [NYC] or Tennessee [TN]), phylogenetic clade (20A, 20C, 19A, 20B, or 19B), and collection date (April 2020, July 2020, or not available [gray]). (B) Schematic representation of the amino acid mutation nomenclature of the variants detected in our cohort. (C) Venn diagram displaying the overlap of the mutations detected in our cohort and the mutations reported on GISAID as of 11 November 2020. (D) Phylogenetic tree generated using Nextclade (version 04-22-21) and modified BioPhylo methods. The phylogeny of a clinical sample pertaining to clade 20C is shown.

**COVID-19 status was associated with an altered metagenomic profile.** To provide a more in-depth profile of other microbes in our samples, 106 metagenomic libraries were sequenced and subsequently processed with MetaPhlAn2. We removed samples with 100% human reads, leading to 22 CVN and 26 CVP samples that were further analyzed for their microbiome composition, AMR profiles, and virulence factors.

We detected a total of 66 bacterial genera and 191 bacterial species within our cohort. Overall, the microbiome profiles identified reflect a normal, commensal nasopharyngeal microbiome, spanning *Streptococcus*, *Veillonella*, *Prevotella*, *Rothia*, *Actinomyces*, *Haemophilus*, and *Neisseria* species (Fig. 4). Additionally, 4 other DNA viruses relevant to human health were detected in the samples, including human herpesvirus 4 and human adenovirus D. Microbial profiles of clinical samples did not cluster based on COVID-19 status.

We next examined sample richness and clustering based on microbiome profiles. Alpha diversity metrics showed no significant difference in richness (observed species and Shannon index) or evenness (Pielou's) between CVP and CVN samples; however, CVN samples exhibited a trend of lower richness (Fig. S1A). Based on the Bray-Curtis principal-coordinate analysis (PCoA) (Fig. S1B), 4 clusters and 5 outliers were observed, of which cluster A and cluster C were largely dominated by *Streptococcus*, *Prevotella*, *Veillonella*, and *Rothia* species, while cluster B and cluster D were dominated by *Prevotella*, *Veillonella*, *Streptococcus*, and *Megasphaera*. Cluster C, with *Streptococcus*/*Veillonella* dominant, and cluster D, with *Prevotella*/*Veillonella* dominant, contained 69.2% of the CVP samples within our

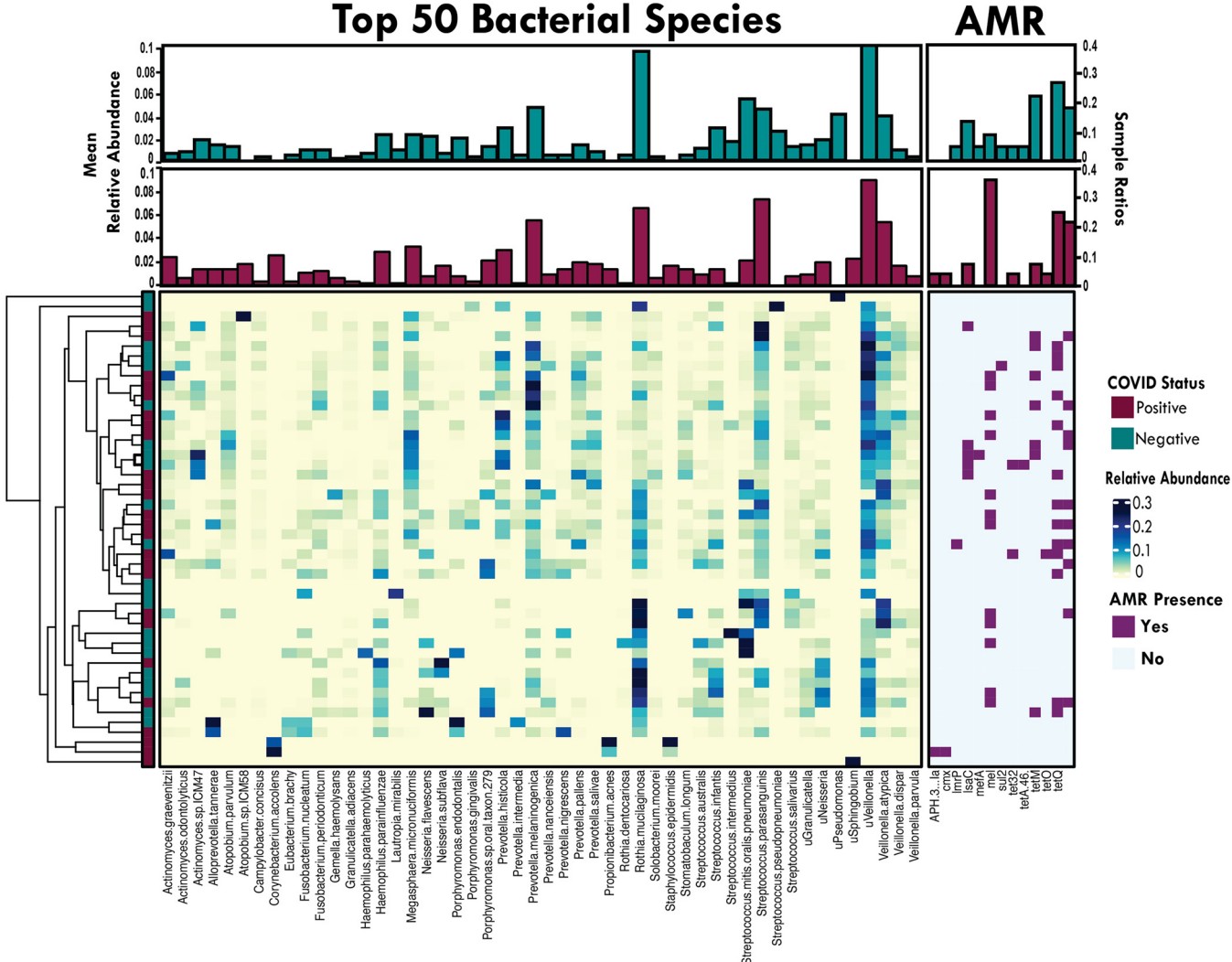

**FIG 4** Nasopharyngeal microbiome profiles. Main heatmap depicts the relative abundances of the top 50 most abundant bacterial species found in our cohort. Top bar plot annotations indicate the mean relative abundance per taxon based on COVID-19 status (CVP [*n* = 26] or CVN [*n* = 22]). Heatmaps display the presence (purple) or absence (light blue) of AMR genes and DNA viruses (green) detected in our samples. A total of 13 different AMR genes were detected across all samples (CVP = 19, CVN = 12, 55 AMR genes total), conferring resistance to several drug classes, including tetracycline [*tet32*, *tetA (46)*, *tetM*, *tetO*, *tetQ*, and *tetW*], macrolides (*mel*, *mefA*, and *lmrP*), aminoglycosides [*APH(3')-la* and *lsaC*], sulfonamides (*sul2*), and phenicols (*cmx*). Top bar plot annotation shows the sample ratios.

cohort. The Bray-Curtis dissimilarity index indicated that *Rothia mucilaginosa* was the main driver of clustering, which was significantly differentiated by the relative abundance levels found by linear discriminant analysis effect size (LEfSe) and Mann-Whitney U test. Moreover, another driver of clustering is explained by the presence/absence of *Streptococcus parasanguinis* (*n* = 18, 12 CVN samples and 6 CVP samples) and unclassified *Neisseria* (*n* = 26, 8 CVN samples and 18 CVP samples), respectively, also indicated by LEfSe and Mann-Whitney U test (data not shown).

Although COVID status did not show clear differences in the NP microbiome communities between CVP and CVN samples, LEfSe revealed that 6 bacterial species and 1 genus (*Actinomyces*) were significantly increased in CVP samples (Fig. 5A), of which 2 species-level taxa, *Actinomyces graevenitzii* and *Prevotella salivae*, were further confirmed when comparing their relative abundances using a Mann-Whitney U test (Fig. 5B), while *Megasphaera micronuciformis*, *Veillonella dispar*, and *Streptococcus gordonii* only exhibited trends of higher relative abundances in CVP samples. Additionally, we found no correlations of bacterial relative abundance versus SARS-CoV-2 $C_T$ values (data not shown).

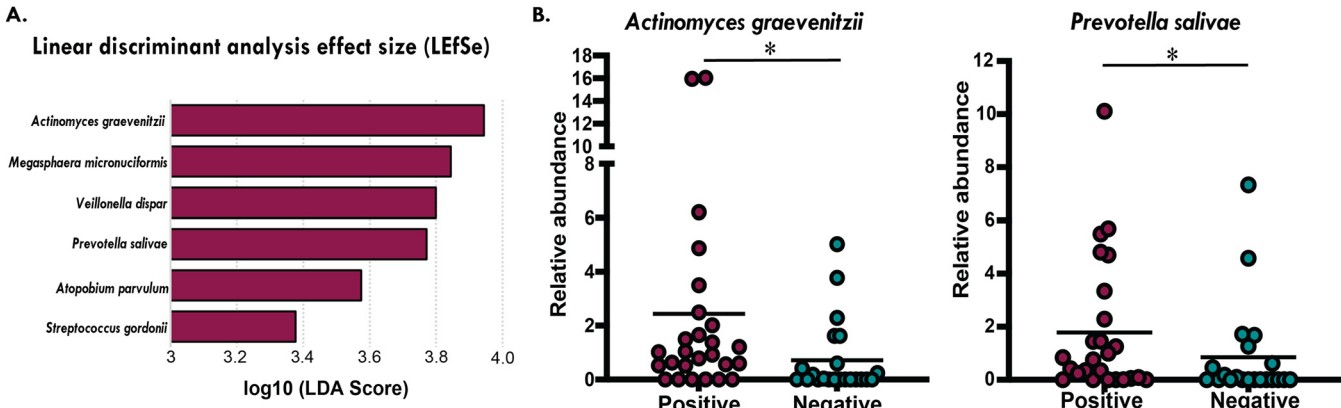

**FIG 5** Nasopharyngeal microbiomes revealed the presence of biomarkers associated with CVP samples. (A) LEfSe analysis showed six bacterial species to be increased in CVP samples. (B) Mann-Whitney U test confirmed significant differences in the relative abundances of *Actinomyces graevenitzii* (*P* = 0.0130) and *Prevotella salivae* (*P* = 0.0394) in CVP samples (*n* = 26) versus CVN samples (*n* = 22).

Furthermore, we compared the classification obtained from MetaPhlAn2 with our *k*-mer-based approach pipeline (Biotia-DX) and found high concordance in the genus-level taxonomy. The top 80% of the most dominant bacterial genera (11/66 total) showed a large overlap (8/11) between the two metagenomic classification tools. Two of the 3 discordant genera were identified in the bottom 20% by each classifier, and only 1 taxon ("*Candidatus* Saccharibacteria") was not detected by MetaPhlAn2 (Fig. S2A).

Since *Prevotella* spp. were found to be a biomarker for CVP status based on the data obtained by two independent bioinformatic pipelines but were discrepant at the species-level classification, we sought to use 16S rRNA gene sequencing to resolve the phylogeny of the correct *Prevotella* species. Sequencing and neighbor-joining phylogenetic analysis of *Prevotella* 16S rRNA gene amplicons confirmed that the biomarker increased in CVP samples is most closely related to *Prevotella salivae* (Fig. S2B).

**Distinct metabolic pathways linked to SARS-CoV-2 positivity.** A total of 434 metabolic pathways were detected in our data set using HUMAnN2, with the most prevalent being those associated with nucleotide, amino acid, lipid, and cell wall biosynthesis. We then compared the differences in metabolic pathways based on COVID-19 status and found 3 metabolic pathways to be overrepresented in CVP samples, while 4 pathways were increased in CVN samples (Fig. 6 and Table S7). Furthermore, we explored the functional profiles associated with the bacterial species in CVP and CVN samples (Table S7).

**Altered antimicrobial resistance and virulence factor profiles in SARS-CoV-2-positive specimens.** For assessing the AMR genes and VFs present in our samples, *k*-mers obtained from the Biotia-DX software were classified using the Comprehensive Antibiotic Resistance Database (CARD) and Virulence Factor Database (VFDB), respectively. A total of 13 different AMR genes were detected across all samples (CVP = 19, CVN = 12, 55 AMR genes total), including genes conferring resistance to several drug classes, such as tetracycline, macrolides, aminoglycosides, sulfonamides, and phenicols (Fig. 4). Interestingly, *mel*, which confers macrolide resistance, was significantly overrepresented in CVP samples. In order to understand which bacteria may be carrying these AMR genes, we correlated the AMR profiles with the bacterial taxa and found *mel* to be significantly correlated with *Streptococcus* spp.

Furthermore, 7 virulence factors were detected in 10 samples, 6 CVN and 4 CVP samples, with the serine-rich repeat protein (SRRP) family (SecA2/SecY2 system) being the most prevalent, found in 3 CVN and 3 CVP samples. Other virulence factors observed included neuraminidase, transferrin-binding protein, direct heme uptake system, major surface protease (MSP), SRRP family, and streptolysin S.

## DISCUSSION

**Novel variant discovery and importance.** Using our novel SARS-CoV-2 NGS assay and COVID-DX pipeline, we found a remarkable number of previously unreported genetic

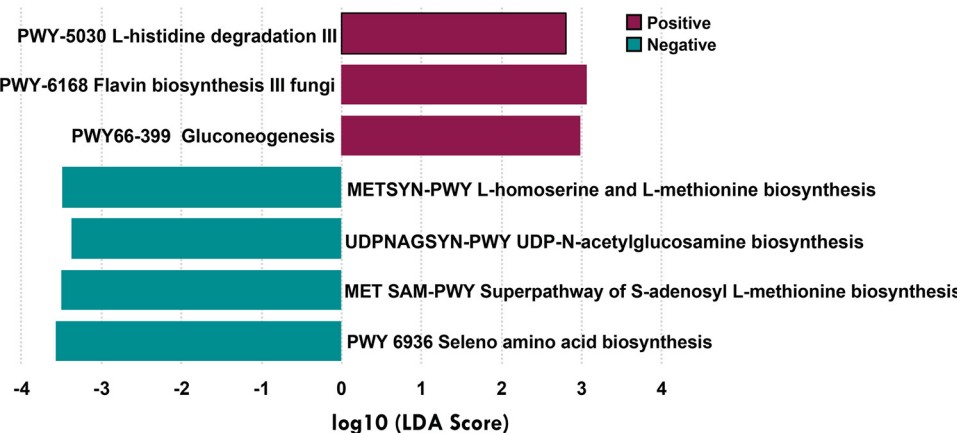

**FIG 6** Functional metagenomic profiles associated with COVID status. We detected 3 pathways (gluconeogenesis [PWY-66-399], flavin biosynthesis [PWY-6168], and L-histidine degradation [PWY-5030]) that were increased in the CVP samples. Four metabolic pathways were found to be overrepresented in CVN samples (L-homoserine and L-methionine biosynthesis, UDP-N-acetylglucosamine biosynthesis, seleno amino acid biosynthesis [PWY-6936], and S-adenosyl L-methionine biosynthesis superpathway). LDA, linear discriminant analysis.

variants, with 55 mutations not described previously, including 21 in the functionally important spike protein. We believe this was facilitated by our hybrid capture NGS-based approach, which enabled robust amplification of the target of interest, generating a greater number of target reads instead of host reads. Our high viral enrichment and genome coverage indicate that the hybrid capture NGS-based approach is a powerful method to detect genetic variants in the SARS-CoV-2 virus; such a method could readily be implemented for other viruses as well.

This test was also found to be sensitive and specific, with a PPA of 96.7% and NPA of 100% compared to the results for a real-time RT-PCR assay, highlighting the utility of a hybrid capture NGS-based approach in the detection and characterization of the SARS-CoV-2 viral genome, especially as we see ongoing evidence of the functional evolution of the virus, warranting widespread and affordable characterization of genetic variants. Our variant findings have been further validated by capturing and sequencing a range of synthetic RNA controls, including the novel strains with larger deletions (6 nucleotides) and single-nucleotide polymorphisms (SNPs) of public health interest (e.g., B.1.1.7). In our clinical cohort, we detected 5 frameshifts after synonymous mutations, of which 2 deletions were not previously reported.

Characterization of genetic variants identified in the SARS-CoV-2 viral genome is important for researchers studying the epidemiology (e.g., transmission and spread) and evolution (e.g., substitution rate and adaptation) of the virus over time. Although the virus appears to have a low, random substitution rate with an average rate of 1 to 2 mutations per month (16), any acquired changes could potentially be selected and alter transmissibility, infectivity, or pathogenicity (17). Mapping variants across the viral genome can also inform drug therapies and guide vaccine development or updates (18). While the majority of mutations have no effect on the fitness of the virus, there is evidence that some variants may make the virus more transmissible (19). Studies are under way to assess potential clinical effects of the existing mutations and whether the lineages observed may make the virus better at evading the host immune response (17, 19, 20). A recent study describes the emerging recurrent deletions in the spike protein that are mapped to antibody epitopes, resulting in resistance to neutralizing antibodies (21).

With limited and low-coverage sequencing, it is possible that variants circulate in a population extensively before detection. Recently, a number of reports described strains just found in the United Kingdom (B.1.1.7 strain) (19, 22), South Africa (B.1.351 strain) (23), Brazil (P.1 strain), and New York, USA (B.1.526) (24). Once described and

tested for specifically, the B.1.1.7 strain was detected in many countries outside the United Kingdom, indicating that this strain may have been circulating and spreading for some time (25). These novel genetic variants are of recent concern because these mutations have been shown to decrease the efficiency and performance of amplicon-based assays, with some tests showing a "dropout" in which one of the genetic targets is not reported (26). Recently, three widely used RT-PCR EUA-authorized diagnostic tests were evaluated due to concerns of a possible increase in the false-positivity rates (26). Furthermore, novel variants have the potential to evade the immune response even after an initial exposure or infection. Although more costly, sufficient sequence coverage is a prerequisite for accurate and sensitive variant calling. More widespread global viral genomic surveillance will be necessary to understand how strain variability impacts vaccine efficacy and long-term immunity. Genetic variant surveillance is also a powerful tool to understand the spread and evolution of the virus to inform public health mitigation efforts of variant-driven outbreaks.

Overall, in line with previous studies, we have found that while the hybrid capture-based workflow is labor intensive and includes more steps and requires higher input amounts than amplicon-based technologies, this approach is highly effective at targeting the total genome and discovering novel variants. It can work with millions of targets per panel and dozens to hundreds of overlapping capture probes, leading to higher specificity for recovering entire viral genomes and demonstrating higher uniformity of coverage compared to amplicon-based technologies (10, 27, 28). While amplicon-based sequencing methods have resulted in discordant single-nucleotide variants, as well as frequently missed variants (10, 29), a capture-based method represents an important tool for addressing missed variants.

**Profiling the SARS-CoV-2 microbiome.** Only a few studies evaluating the microbiome associated with COVID-19 patients have been conducted to date. Overall, our microbial profiling overlapped with previous studies using metagenomic sequencing and analysis of NP specimens (30, 31), detecting commensal respiratory species such as *Streptococcus*, *Veillonella*, *Prevotella*, *Rothia*, *Actinomyces*, *Haemophilus*, and *Neisseria* species. While one study found decreased diversity in SARS-CoV-2-positive specimens, they reported the vast majority of reads mapping to a single organism in each specimen, which could be explained by the necessity for higher coverage of nanopore sequencing (13). We found no significant difference in richness or evenness between CVP and CVN samples, though CVN samples exhibited a trend of lower richness. In addition, Mostafa et al. reported higher abundance of *Corynebacterium accolens* and decreased abundance of *Propionibacteriaceae*, common skin commensals, in CVP samples (13). Although we have not found clear clustering based on SARS-CoV-2 status, our analysis revealed overrepresentation of 7 bacterial taxa in CVP samples, most notably *Actinomyces graevenitzii* and *Prevotella salivae*.

Previous studies have argued that the nasal microbiome composition and dynamics could modulate host immunity and increase susceptibility to upper respiratory infection, particularly acute viral infection (32–34). Interestingly, *Prevotella*, a taxon enriched in CVP patients, has been found to be increased in patients with acute and chronic respiratory diseases, such as influenza and asthma (34–36). The altered microbiome, and specifically, the overrepresentation of *Prevotella* spp., may play a role in modifying the host immunity and, consequently, the host response to viral infections (14). The observed altered species (*Streptococcus*, *Veillonella*, *Prevotella*, and *Rothia* spp.) driving our clusters have been linked to symptomatic and asymptomatic viral respiratory infections in previous studies (37). In concordance with our results, another study on hospitalized COVID-19 patients has reported high prevalences of *Rothia* across patient samples and built environments (38), raising the importance of microbiome profiling in patients with acute viral respiratory infections.

Furthermore, drug resistance profiling is of importance because the COVID-19 pandemic may pose additional infectious disease threats, including a potential rise in AMR. There has been a dramatic increase in the use of antibiotics to treat pneumonia

associated with the viral respiratory infection. One study showed that over 70% of hospitalized COVID-19 patients have received antibiotics throughout their disease course, though only 8% demonstrated bacterial or fungal superinfection (39). Additionally, COVID-19-related long-term hospitalization may lead to increased multidrug-resistant infections and prolonged antibiotic therapy (40). The overrepresentation of macrolide resistance (*melC*) in SARS-CoV-2-positive patients linked to *Streptococcus* spp. may be associated with the underlying viral infection (41). The metabolic profile associated with nucleotide, amino acid, lipid, and cell wall biosynthesis represented by the overrepresentation of different bacterial species may be associated with altered bacterial adherence, host mucosal immunity, and secondary coinfections and can be used for infection biomarkers for risk stratification (42).

In summary, genomic innovations are transforming epidemiology to better characterize, respond to, and prepare for outbreaks. Novel genome-based approaches enable "precision epidemiological responses" and public health interventions on both an individual and population-based scale. Viral metagenomics and the novel hybrid capture NGS-based assay we describe can provide important genetic insights into SARS-CoV-2 and other emerging pathogens and improve surveillance and early detection, potentially preventing or mitigating new outbreaks. Better understanding of the continuously evolving SARS-CoV-2 viral genome and the impact of genetic variants may provide individual risk stratification, precision therapeutic options, improved molecular diagnostics, and population-based therapeutic solutions. Future work includes the extension of this hybrid capture NGS-based assay to characterize other viral and bacterial genomes, total nucleic acid extraction to define the entire microbiome and virome, and the collection of clinical metadata to define risk stratification and disease pathogenesis in relation to SARS-CoV-2 genetic variants. The SARS-CoV-2 pandemic has shown the importance and utility of genomic-variant surveillance as a tool to understand viral dynamics and evolution to guide public health strategies. A hybrid capture approach is applicable not only to SARS-CoV-2 but also to other respiratory pathogens, such as influenza virus, and to future outbreaks.

## MATERIALS AND METHODS

**Specimen collection.** Upper respiratory tract specimens (nasopharyngeal [NP] swabs) were collected following the current CDC guidelines (43) and the manufacturer's instructions for specimen collection. Altogether, 120 specimens from New York City were analyzed using the SARS-CoV-2 NGS assay (validation and independent validation study). An additional 20 SARS-CoV-2-positive specimens from Tennessee were analyzed to define variant differences in samples collected from two geographic locations. Table S1 in the supplemental material contains the specimen collection date, location, collection device, and transport matrix data. For all collections, NP swabs were immediately placed into sterile tubes containing 2 to 3 ml of viral transport medium. Samples were stored for up to 24 h at room temperature or up to 72 h when stored at 2°C to 8°C prior to transportation. After 72 h, all samples were frozen at −70°C or colder until additional testing was performed. The real-time RT-PCR technology that was used to define the presence of SARS-CoV-2 viral RNA is listed in Table S1 and described in the supplemental material. Deidentified samples were collected and processed under an institutional review board (IRB) review numbered Pro00042824 (Advarra).

**RNA and DNA extraction.** RNA and DNA from NP specimens were isolated and purified by manual extraction using the Direct-zol DNA/RNA miniprep kit (250-$\mu$l input volume, spin column, 100-$\mu$g binding capacity; Zymo Research, Irvine, CA). Extracted and purified RNA samples were converted to cDNA through random priming using Random Primer 6, the ProtoScript II first-strand cDNA synthesis kit, and NEBNext Ultra II nondirectional RNA second-strand synthesis kit reagents (NEB, Ipswich, MA).

**Library preparation, target enrichment, and sequencing.** The cDNA samples were converted to Illumina TruSeq-compatible libraries using the Twist library preparation kit with unique dual indices (Twist Bioscience, South San Francisco, CA). After library generation, 8 uniquely barcoded libraries were pooled (187.5 ng per library) to create an 8-plex hybridization reaction. Hybridization was performed for 2 h using the Twist fast hybridization reagents and SARS-CoV-2 research panel, a biotin-bound DNA panel that targets libraries containing the SARS-CoV-2 sequence. These libraries were isolated from human libraries using biotin/streptavidin bead chemistry. Beads were washed several times to reduce the number of off-target libraries in the sequencing pool. All enriched library pools were spiked with 1% PhiX and sequenced on an Illumina NextSeq 550 platform using the NextSeq 500/550 high-output kit (Illumina, San Diego, CA) set to 150-bp single-end reads, which yielded averages of 15.1 million and 1.5 million reads from positive and negative samples, respectively. Picard HsMetrics showed an average of 43% on-target reads, ranging from 0.005% to 99.5%. Additionally, our target enrichment approach yielded an average fold enrichment of 46,791, ranging from 5.9 to 108,602. Subsampling to 500,000

reads per sample did not significantly change fold enrichment (mean of 108,598) or on-target reads (mean of 40.3%). The enrichment efficiency was calculated using the number of reads mapping to targets over the total number of reads in the sample.

**Library preparation and sequencing for metagenomics.** Metagenomic libraries were prepared with the Nextera library preparation kit (Illumina) using 1 to 10 ng of input DNA. After library generation, 12 to 16 uniquely barcoded libraries were pooled and run on an Illumina NextSeq 550 platform using a NextSeq 500/550 mid-output kit (Illumina) set to 150-bp single-end reads, which yielded averages of 10 million and 7.3 million reads from positive and negative samples, respectively. After removal of reads mapping to human DNA, averages of 1.7 million (positive) and 1.9 million (negative) microbial reads were used for further analysis.

**SARS-CoV-2 NGS assay analysis.** The data were analyzed using cloud-based Biotia COVID-DX (version 1.0) software that has been optimized for the SARS-CoV-2 NGS assay. The pipeline software is contained within Docker images orchestrated by Cromwell/WDL (workflow description language) backed by Azure Batch (44). The analysis pipeline included a step to remove low-quality reads, alignment to reference viral and host genomes, extraction of mapped reads, calculation of coverage across the genome, and modeling of coverage to determine the presence/absence of SARS-CoV-2. Furthermore, we detected genomic variants, analyzed phylogenetic and geographic origins, and quantified the SARS-CoV-2 virus. The descriptions of quality and internal controls, inclusivity, exclusivity, and analytical sensitivity studies and the detailed bioinformatic pipeline of the SARS-CoV-2 NGS assay (preprocessing of sequencing data, presence/absence and coverage calculation, genetic variant calculation, and phylogenetic analysis) are included in the supplemental material.

**Clinical metagenomics.** Metagenomics were processed through XSEDE and the Bridges system at the Pittsburgh Supercomputing Center (PSC) (45, 46). Quality analysis was performed with FastQC (version 0.11.3) (47) with default settings to validate the quality of the raw sequencing data. For quality control, adapter trimming and quality filtering were performed using the software fastp (version 0.21.0) (48). Four functions of fastp were used: trimming of autodetected adapter sequences, quality trimming at the 5′ end, quality trimming at the 3′ end, and a screening to detect 512 adapters, including Illumina's Nextera transposase. In order to remove all human reads, Bowtie2 (version 2.3.4.1) (49) and samtools (version 1.9) (50) were used to align the quality-filtered reads to a *Homo sapiens* reference genome (GCF_000001405.39_GRCh38.p13). Quality control with FastQC was performed again to ensure that the unmapped reads obtained were cleaned to run metagenomics tools. Processed reads were analyzed utilizing MetaPhlAn2 (51) to identify the relative abundances of microbial species. An additional *k*-mer-based classification tool (Biotia-DX) was used as an orthogonal bioinformatic validation tool and to improve sensitivity (52). To explore functional genomic profiling, we used HUMAnN2 (53) to characterize microbial pathways linked to the presented microbial species. Antimicrobial resistance and virulence factors were determined using our *k*-mer-based approach matched to the Comprehensive Antibiotic Resistance Database (CARD) (54) and Virulence Factor Database (VFDB) (55). The description of metagenomic validation is in the supplemental material.

**Statistical analysis.** Alpha and beta diversity metrics were calculated with the vegan R package using the relative abundance and count outputs obtained from MetaPhlAn2 and Biotia-DX. The variant heatmap and the microbiome heatmap visualizations with their respective annotations were plotted using the Complex Heatmap R package. All Pearson correlations and correlation plots were calculated and visualized with the Corrplot R package. All other basic statistics and figures were generated with Prism V9.

**Data availability.** All data needed to evaluate conclusions are present in the article, supplemental data and sequencing data were submitted to GISAID.

## SUPPLEMENTAL MATERIAL

Supplemental material is available online only.

**SUPPLEMENTAL FILE 1**, PDF file, 1.2 MB.

## ACKNOWLEDGMENTS

The COVID-DX software pipeline work used CromwellOnAzure, the Microsoft Genomics supported implementation of the Broad Institute's Cromwell workflow engine on Azure. The metagenomic work used the Extreme Science and Engineering Discovery Environment (XSEDE), which is supported by National Science Foundation grant (ACI-1548562). Specifically, it used the Bridges system, which is supported by NSF award (ACI-1445606), at the Pittsburgh Supercomputing Center (PSC). This work used resources of the COVID-19 HPC Consortium. We thank Nicholas Nystrom, Paola Buitrago, Julian Uran, David O'Neal and collaborators for their assistance with PSC resources access, use, and software installation support. We also thank Zaineb Bello and Caitlin Otto for their laboratory operation support; the Abesse Team, Pierre Davidoff, Shay David and Cory Mason for IT support; Peter Eugster and Bryan Hoglund for logistical support; Shakil Ahmed for regulatory support; and Bradley Connor for his clinical advice and reporting support.

D.N.-S., M.C.-R., C.E.M., and N.B.O. designed the study. K.B. and S.C. designed the SARS-CoV-2 NGS assay; D.N.-S., J.E.B., H.L.W., M.D., C.E.M., N.B.O. designed and validated

the analysis software tool; D.N.-S., M.C.-R., J.E.B., performed the clinical validation; R.J.B., M.K.T. and C.B.J. provided the clinical samples and orthogonal analysis; A.B. performed variant curation; D.N.-S., M.C.-R., and M.D. designed and performed the metagenomic analysis. D.N.-S., M.C.-R., C.H. and N.B.O. wrote the manuscript with input from all authors.

The authors declare that D.N.-S., M.C.-R., J.E.B., H.L.W., M.D., A.B., C.H., C.E.M., and N.B.O. are employees of Biotia, Inc. K.B. and S.C. are employees of Twist Bioscience.

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
