## [Reviewer comments · Microbiology Spectrum]

Microbiology Spectrum

Targeted Hybridization Capture of SARS-CoV-2 and Metagenomics Enables Genetic Variant Discovery and Nasal Microbiome Insights

Dorottya Nagy-Szakal, Mara Couto-Rodriguez, Heather Wells, Joseph Barrows, Marilyne Debieu, Kristin Butcher, Siyuan Chen, Agnes Berki, Courteny Hager, Robert Boorstein, Mariah Taylor, Colleen Jonsson, Christopher Mason, and Niamh O'Hara

Corresponding Author(s): Niamh O'Hara, Biotia

Review Timeline:

Submission Date:	May 6, 2021
Editorial Decision:	June 22, 2021
Revision Received:	July 8, 2021
Accepted:	July 21, 2021

Editor: Heba Mostafa

Reviewer(s): The reviewers have opted to remain anonymous.

Transaction Report:

DOI: <https://doi.org/10.1128/Spectrum.00197-21>

June 22, 2021

Dr. Niamh B. O'Hara
Biotia
New York

Re: Spectrum00197-21 (Targeted Hybridization Capture of SARS-CoV-2 and Metagenomics Enables Genetic Variant Discovery and Nasal Microbiome Insights)

Dear Dr. Niamh B. O'Hara:

Thank you for submitting your manuscript to Microbiology Spectrum. When submitting the revised version of your paper, please provide (1) point-by-point responses to the issues raised by the reviewers as file type "Response to Reviewers," not in your cover letter, and (2) a PDF file that indicates the changes from the original submission (by highlighting or underlining the changes) as file type "Marked Up Manuscript - For Review Only". Please use this link to submit your revised manuscript - we strongly recommend that you submit your paper within the next 60 days or reach out to me. Detailed information on submitting your revised paper are below.

Link Not Available

Sincerely,

Heba Mostafa

Journals Department
Reviewer comments:

Reviewer #1 (Comments for the Author):

This study by Nagy-Szakala, D., Couto-Rodriguez, M., et al describes a hybridization capture NGS assay for identification and sequencing of clinical isolates of SARS-CoV-2. The developed assay is highly sensitive, comparable in reliability to traditional RT-PCR and importantly, can identify SARS-CoV-2 genetic variants. In addition, the NGS pipeline was utilized to profile the microbiome isolated from specimen. The coupling of SARS-CoV-2 detection and sequencing will provide a valuable tool in improving the surveillance of SARS-CoV-2 variants as well as identification of potentially new variants of concern. The role of bacteria during COVID-19 has been debated and evidence for and against have been published since the onset of the pandemic. Here, the authors identify a probable relationship between certain taxa of bacteria and SARS-CoV-2 infection. Overall, the methods and data presented in this study will be an important addition to the current SARS-CoV-2 surveillance toolkit. Listed below are some comments that will further broaden the application of the presented assay.

1. The authors observe an enrichment of certain bacterial taxa in SARS-CoV-2 positive specimens. Generating a correlation plot between the observed bacterial taxa and SARS-CoV-2 Ct values will provide a rationale for proposing a role of the specific bacteria during SARS-CoV-2 infection. Furthermore, an additional correlation between clinical score and bacteria should be presented.
2. Shotgun metagenomics was performed on DNA extracted from samples. Most respiratory viruses that impact global health are RNA viruses and would not be assessed by the current pipeline. This downfall of the current assay should be discussed and improved upon. The presence of influenza virus, a negative sense RNA virus, in Figure 4 should be explained.
3. Several studies have demonstrated the potential of animal reservoirs of SARS-CoV-2 such as cats, dogs, mink among others. Surveillance of animal reservoirs using the presented assay will not only add to our understanding of SARS-CoV-2 adaptation during infection of animal hosts but could potentially identify viruses similar to SARS-CoV-2 with pandemic potential.
4. Seasonal respiratory viruses have a major yearly burden on human health. Adapting the current pipeline for surveillance of Influenza viruses should be discussed.
5. In Figure 3D, the dot color used to represent "TEST Sample" and "19A" are very similar and should be changed to make them clearly distinguished.
6. Figure 5C and 5D are out of date and need to be updated to include the most recent genome sequence and phylogenetic clades, respectively.
7. In Figure 5B (*Actinomyces graevenitzii*), there are two clear outliers present in the positive samples. These specific samples should be discussed in the context of SARS-CoV-2 and presence of other bacteria taxa.

Reviewer #2 (Comments for the Author):

The manuscript by Nagy-Szakal et al describes the development and validation of a new NGS method for sequencing SARS-CoV-2 genome. This new method depends on targeted capture and sequencing the genome of the COVID19-causing virus. The manuscript also describes a novel pipeline to analyze NGS data using the capture hybridization method. In addition, the authors profiled the metagenomics and the microbiome of nasopharyngeal samples from COVID-19 positive and negative cases.

Generally, the manuscript is well-written and potentially impactful. However, I have the following comments that are needed to be addressed before publishing this paper:

- The authors claims that amplicon-based sequencing may fail to detect novel SARS-CoV-2

variants. While this is a hypothetical possibility, the authors provide no evidence that this is the case. Indeed, the positive and negative percent agreements for detection of SARS-CoV-2 with standard RT-PCR are 97 and 100%, respectively. These data suggest that amplicon-based sequencing can detect different variants seen in this study

- For metagenomic data, it is important to stress that SARS-CoV-2 negative samples should not be considered normal as many of these samples are likely collected from cases with symptoms suggesting COVID19, so many of these cases may have other respiratory diseases

Staff Comments:

Preparing Revision Guidelines

For complete guidelines on revision requirements, please see the Instructions to Authors at [link to page]. **Submissions of a paper that does not conform to Microbiology Spectrum guidelines will delay acceptance of your manuscript.**

Please return the manuscript within 60 days; if you cannot complete the modification within this time period, please contact me. If you do not wish to modify the manuscript and prefer to submit it to another journal, please notify me of your decision immediately so that the manuscript may be formally withdrawn from consideration by Microbiology Spectrum.

If you would like to submit an image for consideration as the Featured Image for an issue, please contact Spectrum staff.

This study by Nagy-Szakala, D., Couto-Rodriguez, M., *et al* describes a hybridization capture NGS assay for identification and sequencing of clinical isolates of SARS-CoV-2. The developed assay is highly sensitive, comparable in reliability to traditional RT-PCR and importantly, can identify SARS-CoV-2 genetic variants. In addition, the NGS pipeline was utilized to profile the microbiome isolated from specimen. The coupling of SARS-CoV-2 detection and sequencing will provide a valuable tool in improving the surveillance of SARS-CoV-2 variants as well as identification of potentially new variants of concern. The role of bacteria during COVID-19 has been debated and evidence for and against have been published since the onset of the pandemic. Here, the authors identify a probable relationship between certain taxa of bacteria and SARS-CoV-2 infection. Overall, the methods and data presented in this study will be an important addition to the current SARS-CoV-2 surveillance toolkit. Listed below are some comments that will further broaden the application of the presented assay.

1. The authors observe an enrichment of certain bacterial taxa in SARS-CoV-2 positive specimens. Generating a correlation plot between the observed bacterial taxa and SARS-CoV-2 Ct values will provide a rationale for proposing a role of the specific bacteria during SARS-CoV-2 infection. Furthermore, an additional correlation between clinical score and bacteria should be presented.
2. Shotgun metagenomics was performed on DNA extracted from samples. Most respiratory viruses that impact global health are RNA viruses and would not be assessed by the current pipeline. This downfall of the current assay should be discussed and improved upon. The presence of influenza virus, a negative sense RNA virus, in Figure 4 should be explained.
3. Several studies have demonstrated the potential of animal reservoirs of SARS-CoV-2 such as cats, dogs, mink among others. Surveillance of animal reservoirs using the presented assay will not only add to our understanding of SARS-CoV-2 adaptation during infection of animal hosts but could potentially identify viruses similar to SARS-CoV-2 with pandemic potential.
4. Seasonal respiratory viruses have a major yearly burden on human health. Adapting the current pipeline for surveillance of Influenza viruses should be discussed.
5. In Figure 3D, the dot color used to represent "TEST Sample" and "19A" are very similar and should be changed to make them clearly distinguished.
6. Figure 5C and 5D are out of date and need to be updated to include the most recent genome sequence and phylogenetic clades, respectively.
7. In Figure 5B (*Actinomyces graevenitzi*), there are two clear outliers present in the positive samples. These specific samples should be discussed in the context of SARS-CoV-2 and presence of other bacteria taxa.

**Response to Reviewers
Spectrum 00197-21**

Dr. Heba Mustafa
Editor | Microbiology Spectrum

RE: Manuscript submission #00197-21

Dear Dr. Heba Mustafa and Reviewers,

Enclosed please find a response to reviewers' comments on manuscript #00197-21 as well as the revised manuscript. The authors of this manuscript would like to thank the reviewers for their insightful questions and comments. We have addressed their comments below and made the necessary modifications to the manuscript, which are highlighted in yellow.

Best wishes,

Dr. Niamh O'Hara

Reviewer comments:

Reviewer #1 (Comments for the Author):

This study by Nagy-Szakal, D., Couto-Rodriguez, M., et al describes a hybridization capture NGS assay for identification and sequencing of clinical isolates of SARS-CoV-2. The developed assay is highly sensitive, comparable in reliability to traditional RT-PCR and importantly, can identify SARS-CoV-2 genetic variants. In addition, the NGS pipeline was utilized to profile the microbiome isolated from specimen. The coupling of SARS-CoV-2 detection and sequencing will provide a valuable tool in improving the surveillance of SARS-CoV-2 variants as well as identification of potentially new variants of concern. The role of bacteria during COVID-19 has been debated and evidence for and against have been published since the onset of the pandemic. Here, the authors identify a probable relationship between certain taxa of bacteria and SARS-CoV-2 infection. Overall, the methods and data presented in this study will be an important addition to the current SARS-CoV-2 surveillance toolkit. Listed below are some comments that will further broaden the application of the presented assay.

1. The authors observe an enrichment of certain bacterial taxa in SARS-CoV-2 positive specimens. Generating a correlation plot between the observed bacterial taxa and SARS-

CoV-2 Ct values will provide a rationale for proposing a role of the specific bacteria during SARS-CoV-2 infection. Furthermore, an additional correlation between clinical score and bacteria should be presented.

Response: Thank you for that feedback. In response, we generated a Pearson correlation plot between the top 50 most abundant bacterial taxa and SARS-CoV-2 Ct values. However, we observed no correlation among these variables. The enrichment observed and outlined in the paper was determined by comparing the SARS-CoV-2 infection vs no SARS-CoV-2 infection states. It is possible that viral load does not play a role in the enrichment observed, or that our data is not appropriate to fully address this.

2. Shotgun metagenomics was performed on DNA extracted from samples. Most respiratory viruses that impact global health are RNA viruses and would not be assessed by the current pipeline. This downfall of the current assay should be discussed and improved upon. The presence of influenza virus, a negative sense RNA virus, in Figure 4 should be explained.

Response: We extended our discussion including other RNA and DNA viruses to the hybrid-capture based assay on P. 20 including “Future work includes the extension of this hybrid capture NGS-based assay to characterize other viral and bacterial genomes, *total nucleic acid extraction to define the entire microbiome and virome*, and the collection of clinical metadata to define risk stratification and disease pathogenesis in relation to SARS-CoV-2

genetic variants". Also, using total nucleic acid instead of DNA only could provide a deeper insight into not only the microbiome, but the entire virome, including RNA viruses.

Regarding influenza, with your feedback and our subsequent analysis we have determined this to be an accidental detection of RNA viruses (such as influenza A). This was further investigated using BLAST to map to the influenza and other viruses, it turned out to be a false positive. We removed all RNA viruses from the manuscript text and also updated Figure 4 removing the viral information.

3. Several studies have demonstrated the potential of animal reservoirs of SARS-CoV-2 such as cats, dogs, mink among others. Surveillance of animal reservoirs using the presented assay will not only add to our understanding of SARS-CoV-2 adaptation during infection of animal hosts but could potentially identify viruses similar to SARS-CoV-2 with pandemic potential.

Response: We agree this context is important. We added a section explaining the keys of using this technology for surveillance of animal reservoirs to Prospective (Supplementary Materials).

"NGS provides a valuable tool for the detection of emerging viruses in domestic animals and wildlife, and generates critical data that is needed to characterize the potential for a virus to be pathogenic in humans. Several studies have demonstrated the potential of animal reservoirs of SARS-CoV-2 such as cats, dogs, bats, minks among others (6). Surveillance of animal reservoirs using the presented assay will not only add to our understanding of SARS-CoV-2 adaptation during infection of animal hosts but could potentially identify viruses similar to SARS-CoV-2 with pandemic potential."

4. Seasonal respiratory viruses have a major yearly burden on human health. Adapting the current pipeline for surveillance of Influenza viruses should be discussed.

Response: We extended the discussion with additional focus on using hybrid-capture based NGS assays determining other respiratory pathogens and influenza.

"Future work includes the extension of this hybrid capture NGS-based assay to characterize other viral and bacterial genomes, *total nucleic acid extraction to define the entire microbiome and virome*, and the collection of clinical metadata to define risk stratification and disease pathogenesis in relation to SARS-CoV-2 genetic variants. *The SARS-CoV-2 pandemic has shown the importance and utility of genomic variant surveillance as a tool to understand viral dynamics and evolution to guide public health strategies. A hybrid capture approach is not only applicable to SARS-CoV-2 but also for other respiratory pathogens such as influenza and future outbreaks.*"

5. In Figure 3D, the dot color used to represent "TEST Sample" and "19A" are very similar and should be changed to make them clearly distinguished.

Response: We updated the phylogenetic tree and now the test sample shows clearly the clade it belongs to (see also comments for next questions).

6. Figure 5C and 5D are out of date and need to be updated to include the most recent genome sequence and phylogenetic clades, respectively.

Response: We believe the reviewer requested the update of Figure 3C and 3D. We updated the phylogenetic analysis (Command-line Nextclade (04/22/2021) was used to identify the clade and generate a phylogenetic tree. The tree was displayed using a modified version of BioPhylo). We also provided updated information on new vs. reported variants (Supplementary table S5). As of 06/23/2021, we found 38 new and 141 reported mutations in our validation cohort.

7. In Figure 5B (*Actinomyces graevenitzii*), there are two clear outliers present in the positive samples. These specific samples should be discussed in the context of SARS-CoV-2 and presence of other bacteria taxa.

Response: There are two CVP samples that have a higher relative abundance of *Actinomyces graevenitzii* (~16%) than other CVP samples (mean: 2.4%). The mean observed species for all CVP samples was 48, while these samples had an observed species of 48 and 60. One of the samples had lower relative abundance of *Rothia mucilaginosa* (0.4%) while the other sample had lower levels of *Veillonella atypica* (0.3%) than other CVP samples.

Reviewer #2 (Comments for the Author):

The manuscript by Nagy-Szakal et al describes the development and validation of a new NGS method for sequencing SARS-CoV-2 genome. This new method depends on targeted capture and sequencing the genome of the COVID19-causing virus. The manuscript also describes a novel pipeline to analyze NGS data using the capture hybridization method. In addition, the authors profiled the metagenomics and the microbiome of nasopharyngeal samples from COVID-19 positive and negative cases.

Generally, the manuscript is well-written and potentially impactful. However, I have the following comments that are needed to be addressed before publishing this paper:

1. The authors claims that amplicon-based sequencing may fail to detect novel SARS-CoV-2 variants. While this is a hypothetical possibility, the authors provide no evidence that this is the case. Indeed, the positive and negative percent agreements for detection of SARS-CoV-2 with standard RT-PCR are 97 and 100%, respectively. These data suggest that amplicon-based sequencing can detect different variants seen in this study

Response: Thank you for this feedback, we are happy to provide additional context. While RT-PCR is a widely used diagnostic assay because of its scalability and sensitivity in determining presence/absence of a pathogen, it only targets 1-3 small fragments (~100-200bp) along the viral genome, while amplicon-based NGS approaches, such as ARTIC, tile across the whole viral genome generating 400bp amplicons of ~90 targets. We used an RT-PCR as a comparator assay to determine presence/absence (because these are more widely accepted by the FDA) and not an amplicon-based sequencing approach. In regards to variant detection, it is widely reported in the literature that a hybrid capture sequencing approach yields more even coverage of the genome, tolerates higher number of mismatches (20%) due to the probe length (120bp dsDNA probes) and is more suited for new variants discovery (WHO: Genomic sequencing of SARS-CoV-2: a guide to implementation for maximum impact on public health; <https://www.who.int/publications/i/item/9789240018440>, cited in the paper for context). Amplicon-based approaches exhibit primer dropouts due to lower tolerance to mismatches on the primer binding sites.

2. For metagenomic data, it is important to stress that SARS-CoV-2 negative samples should not be considered normal as many of these samples are likely collected from cases with symptoms suggesting COVID19, so many of these cases may have other respiratory diseases

Response: We appreciate the reviewer comment, and indeed, it is a valid concern that these samples were taken from symptomatic individuals during the surveillance effort in the beginning of the pandemic. Since we did not have access to the clinical metadata for this study, we added this concern to our Limitation Section (Supplementary Materials). “SARS-CoV-2 negative samples should not be considered healthy since clinical metadata related to any respiratory symptoms have not been reported at the time of collection.”

July 21, 2021

Dr. Niamh B. O'Hara
Biotia
New York

Re: Spectrum00197-21R1 (Targeted Hybridization Capture of SARS-CoV-2 and Metagenomics Enables Genetic Variant Discovery and Nasal Microbiome Insights)

Dear Dr. Niamh B. O'Hara:

Your manuscript has been accepted, and I am forwarding it to the ASM Journals Department for publication. You will be notified when your proofs are ready to be viewed.

Sincerely,

Heba Mostafa
Editor, Microbiology Spectrum
